# A male-killing gene encoded by a symbiotic virus of *Drosophila*

Daisuke Kageyama [1,9] ✉, Toshiyuki Harumoto [2,3,9], Keisuke Nagamine[1], Akiko Fujiwara [4,5], Takafumi N. Sugimoto[1], Akiya Jouraku[1], Masaru Tamura[6], Takehiro K. Katoh[7,8] & Masayoshi Watada [7,8] ✉

In most eukaryotes, biparentally inherited nuclear genomes and maternally inherited cytoplasmic genomes have different evolutionary interests. Strongly female-biased sex ratios that are repeatedly observed in various arthropods often result from the male-specific lethality (male-killing) induced by maternally inherited symbiotic bacteria such as *Spiroplasma* and *Wolbachia*. However, despite some plausible case reports wherein viruses are raised as male-killers, it is not well understood how viruses, having much smaller genomes than bacteria, are capable of inducing male-killing. Here we show that a maternally inherited double-stranded RNA (dsRNA) virus belonging to the family Partitiviridae (designated DbMKPV1) induces male-killing in *Drosophila*. DbMKPV1 localizes in the cytoplasm and possesses only four genes, i.e., one gene in each of the four genomic segments (dsRNA1–dsRNA4), in contrast to ca. 1000 or more genes possessed by *Spiroplasma* or *Wolbachia*. We also show that a protein (designated PVMKp1; 330 amino acids in size), encoded by a gene on the dsRNA4 segment, is necessary and sufficient for inducing male-killing. Our results imply that male-killing genes can be easily acquired by symbiotic viruses through reassortment and that symbiotic viruses are hidden players in arthropod evolution. We anticipate that host-manipulating genes possessed by symbiotic viruses can be utilized for controlling arthropods.

In most eukaryotes, biparentally inherited nuclear genomes and maternally inherited cytoplasmic genomes have different evolutionary interests[1,2]. A female-biased sex ratio is advantageous for cytoplasmic genomes while a 1:1 sex ratio is adaptive for nuclear (autosomal) genomes; thus, there exists a conflict over the sex ratio. In arthropods, strongly female-biased sex ratios have been repeatedly observed in various arthropods, which resulted from the male-specific lethality

(male-killing) induced by maternally inherited symbiotic bacteria such as *Spiroplasma* and *Wolbachia*[3–5]. Likewise, heritable viruses in insects are known but are generally thought to be uncommon (e.g., sigma virus[6]). Such viruses are classically biparentally inherited and thus are not expected to manipulate host reproduction. Recent studies investigating the metagenomics of various arthropod taxa have unearthed a wide array of viruses[7], some of which are likely to be vertically

[1]Institute of Agrobiological Sciences, National Agriculture and Food Research Organization, 1-2 Owashi, Tsukuba, Ibaraki 305-0851, Japan. [2]Hakubi Center for Advanced Research, Kyoto University, Yoshida-honmachi, Sakyo-ku, Kyoto 606-8501, Japan. [3]Graduate School of Biostudies, Kyoto University, Yoshida-Konoe-cho, Sakyo-ku, Kyoto 606-8501, Japan. [4]Center for Food Science and Wellness, Gunma University, 4-2 Aramaki, Maebashi, Gunma 371-8510, Japan. [5]Chemical Genomics Research Group, RIKEN Center for Sustainable Resource Science, Wako, Saitama 351-0198, Japan. [6]Division of Food Safety Information, National Institute of Health Sciences, 3-25-26 Tonomachi, Kawasaki-ku, Kawasaki, Kanagawa 210-9501, Japan. [7]Graduate School of Science and Engineering, Ehime University, Matsuyama, Ehime 780-8857, Japan. [8]Present address: Department of Biological Sciences, Tokyo Metropolitan University, 1-1 Minamiosawa, Hachioji, Tokyo 192-0397, Japan. [9]These authors contributed equally: Daisuke Kageyama, Toshiyuki Harumoto. ✉e-mail: kagymad@affrc.go.jp; watada4123@gmail.com

transmitted endosymbionts, and their effects on host ecology as well as evolution are gaining attention[8-11].

Previously, we reported an all-female matriline (SP12F) in *Drosophila biauraria*[5]. We observed a lack of polymerase chain reaction (PCR) amplicons after the use of bacterial universal primers, the lack of an effect of antibiotic treatment on the sex ratio, transmissibility of the all-female trait after injecting fly homogenate sterilized using a 0.22-μm filter, and matrilineal inheritance of the all-female trait. These features strongly suggested that a non-bacterial agent—possibly a virus—was responsible for the absence of males[5]. Males of the all-female matriline were considered to be killed during embryogenesis because the egg hatch rate was nearly half of that of a normal sex ratio line (SP11-20) despite there being no significant differences in the total number of eggs laid between the two lines[5]. This phenomenon, also known as male-killing, is one of the reproductive manipulations commonly exerted by endosymbiotic bacteria, such as *Wolbachia*[4]. However, the above-mentioned observations led us to suspect the incidence of virus-induced male-killing.

Here we show that the cause of male-killing is a virus belonging to the family Partitiviridae, reveal the genome structure of the virus and identify a gene responsible for male-killing.

## Results and discussion
### Male killing during late embryogenesis
We confirmed that egg hatch rates of the all-female matrilines, SP12F (the originally discovered all-female matriline) and tr.SP11-20 (the matriline generated by injecting SP12F homogenate into SP11-20)[5], were nearly half of those of SP11-20, which corresponds to a typical feature of early male-killing (Fig. 1A). Four days after oviposition, well-developed but dead embryos (e.g., with black mouth parts and a differentiated cuticle) were abundant in all-female matrilines (SP12F and tr.SP11-20), but not in the normal line (SP11-20) (Supplementary Fig. 1). PCR with Y chromosome-specific primers showed that, in SP12F, 93.3–96.6% (median: 93.5%) of the dead embryos were XY (i.e., males),

while only 0–2.2% (median: 0%) of the hatched larvae were XY (Fig. 1B). Similarly, in tr.SP11-20, 95.7–97.8% (median: 97.8%) of the dead embryos were XY, while 3.5–34.8% (median: 8.7%) of the hatched larvae were XY. In the control line (SP11-20), 50.0–58.3% (median: 50%) of the hatched larvae were XY. These data suggested that the low egg hatch rates of the SP12F and tr.SP11-20 were predominantly caused by the high mortality of male embryos during late embryogenesis. We also observed that a few SP12F/tr.SP11-20 males that succeeded in hatching from eggs died before adulthood.

### Identification of RNAs associated with male-killing
To identify the causal non-bacterial agent(s) responsible for male-killing, we compared the RNA sequencing data of seven lines of *D. biauraria*—two matrilines that show all-female trait (SP12F and tr.SP11-20), and five genetically independent lines with nearly 1:1 "normal" sex ratio (SP11-20, TM15-12, TM15-22, TM15-41, and TM15-47) (Supplementary Fig. 2). Among the 120,354 total contigs, four contigs (DN1742_c0_g1, DN8358_c0_g1, DN17116_c0_g2, and DN18525_c0_g4) showed a sharp contrast between all-female matrilines and normal sex ratio matrilines (Supplementary Fig. 3). In each of the two all-female matrilines, the four contigs were among the 40 contigs that showed largest numbers of reads (top 0.0332%) (transcripts per million (TPM): 2767.31–3951.09 in SP12F and 2061.94–3043.84 in tr.SP11-20), while they showed zero or small numbers of reads (TPM < 0.2) in normal sex ratio matrilines (Supplementary Fig. 4). A contig (DN17987_c0_g2_i4) that ranked 89th based on the number of reads also showed a sharp contrast between fly lines, but not in accordance with the sex ratio trait. This contig was homologous to that of *Gluconacetobacter* sp., a gut symbiont of *Drosophila*[12].

### Partiti-like virus as the cause of male-killing
For each of the four contigs, both ends of the transcripts were successfully determined through rapid amplification of cDNA 3′-ends (3′-RACE) using sense and antisense primers (Supplementary Table 1), indicating that they consist of double-stranded RNA (dsRNA). Each of

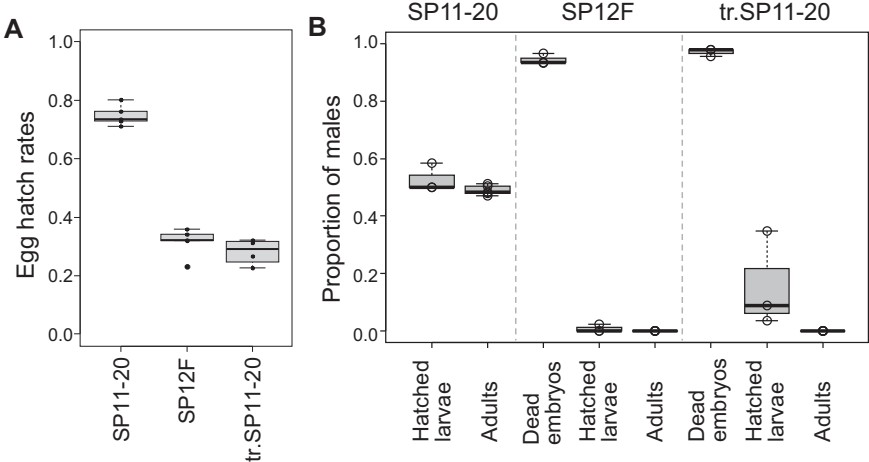

**Fig. 1 | Male-killing occurs during late embryogenesis. A** Egg hatch rates of a normal line (SP11-20), an all-female matriline (SP12F), and a transinfected line (tr.SP11-20) of *D. biauraria*. Box plots represent the median, the first quartiles and third quartiles with whiskers drawn within the 1.5 IQR value. Points outside the whiskers are outliers. SP11-20: *n* = 180, 152, 135, 373, and 879 eggs examined over 5 independent experiments. SP12F: *n* = 120, 135, 199, 217, and 810 eggs examined over 5 independent experiments. tr.SP11-20: *n* = 310, 754, 689, and 866 eggs examined over 4 independent experiments. **B** Proportion of males of dead (but well-developed) embryos, hatched larvae, and adults in SP11-20, SP12F, and tr.SP11-20 (For SP11-20, dead embryos were excluded from the analysis). Box plots represent the median, the first quartiles and third quartiles with whiskers drawn within the 1.5 IQR value. Points outside the whiskers are outliers. Hatched larvae

(SP11-20): *n* = 60, 46, and 46 individuals examined over 3 independent experiments. Emerged adults (SP11-20): *n* = 127, 111, 130, 83, and 98 individuals examined over 5 independent experiments. Unhatched dead embryos (SP12F): *n* = 59, 46, and 45 individuals examined over 3 independent experiments. Hatched larvae (SP12F): *n* = 65, 43, and 46 individuals examined over 3 independent experiments. Emerged adults (SP12F): *n* = 217, 139, 82, 67, and 33 individuals examined over 5 independent experiments. Unhatched dead embryos (tr.SP11-20): *n* = 46, 46, and 46 individuals examined over 3 independent experiments. Hatched larvae (tr.SP11-20): *n* = 57, 46, and 46 individuals examined over 3 independent experiments. Emerged adults (tr.SP11-20): *n* = 88, 60, 87, 80, and 47 individuals examined over 5 independent experiments. Embryos and larvae were sexed based on genomic PCR using Y chromosome-specific primers. Raw data are provided in Supplementary Data 1.

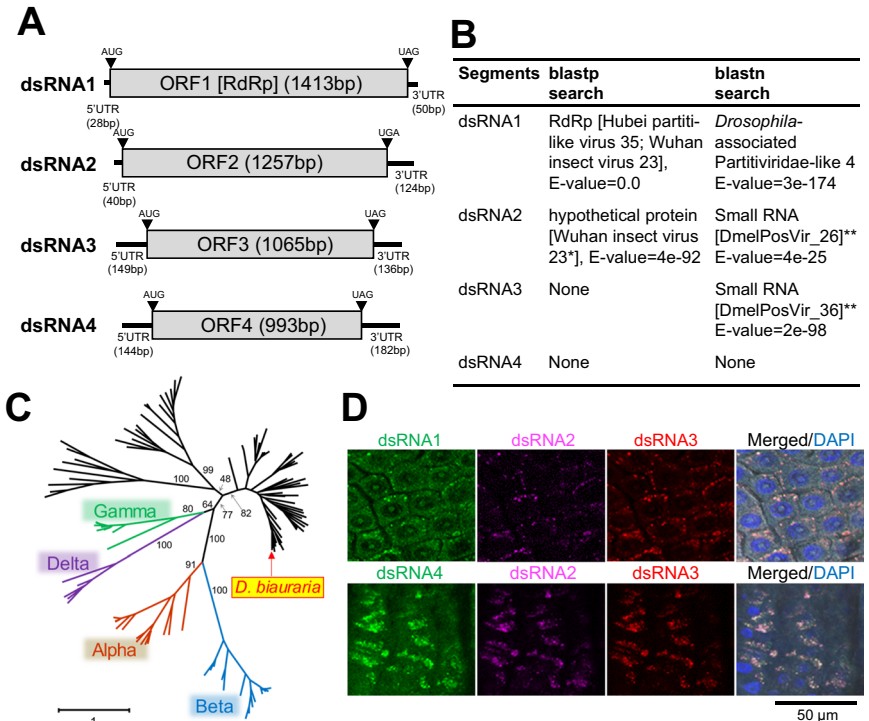

**Fig. 2 | Causal agent of the male-killing. A** Organization of four dsRNA sequences (dsRNA1–dsRNA4) identified in SP12F and tr.SP11-20 lines. Gray boxes: putative open reading frames (ORFs). Bars: putative untranslated regions (5′ UTR and 3′ UTR). Black triangles: start codons (AUG) and stop codons (UAG and UGA). **B** Results of database search for the four dsRNAs. *Considered Partitiviridae because the other segment of the Wuhan insect virus 23 is homologous to Partitiviridae RdRp (Shi et al., 2016). **Putative viral sequences identified in *Drosophila* (Webster et al., 2015). **C** Unrooted phylogenetic tree of family Partitiviridae based on amino acid sequences of RNA-dependent RNA polymerase (RdRp). Colored branches indicate known partitiviruses (Alpha: Alphapartitivirus, Beta: Betapartitivirus, Gamma: Gammapartitivirus, Delta: Deltapartitivirus). Black branches indicate Partiti-like virus sequences deposited as transcriptome shotgun assembly from various arthropod species. Red arrow indicates the dsRNA1 (ORF1) identified in *D. biauraria* SP12F. Amino acid sequences of RdRp and host information of the viruses used for phylogenetic analysis are provided in Supplementary Data 2. **D** In vitro localization of the four RNA sequences (dsRNA1–dsRNA4). Upper: adult midgut tissue hybridized to the fluorescent probes of dsRNA1, dsRNA2, and dsRNA3 (green, magenta, and red, respectively), and counterstained with 4′,6-diamidino-2-phenylindole (DAPI) (blue) overlaid with DIC (differential interference contrast images) (*n* = 9). Bottom: adult midgut tissue hybridized to the fluorescent probes of dsRNA4, dsRNA2, and dsRNA3 (green, magenta, and red, respectively) and counterstained with DAPI (blue) overlaid with DIC (*n* = 9). Original photos are provided as Supplementary Fig. 8.

these dsRNAs, referred to as dsRNA1 (1491 bp) [LC704637], dsRNA2 (1421 bp) [LC704638], dsRNA3 (1350 bp) [LC704639], and dsRNA4 (1319 bp) [LC704640] was predicted to contain a single open reading frame (ORF) (ORF1–ORF4, respectively) (Fig. 2A). The amino acid sequence of ORF1 was homologous to RNA-dependent RNA polymerase (RdRp) of Partitiviridae-like dsRNA viruses (Fig. 2B; Supplementary Fig. 5; Supplementary Table 2). Phylogenetic analyses based on amino acid sequences demonstrated that ORF1 belongs to a clade including *Gammapartitivirus* (fungal viruses) and Partiti-like virus sequences deposited as transcriptome shotgun assembly from various arthropod species (Fig. 2C). ORF1 was most closely related to a sequence previously identified in wild-caught *Drosophila* (KP757931; *Drosophila*-associated Partitivirus-like sequence 4 derived from a large pool of *D. ananassae, D. melanogaster, D. malerkotliana*, and *Scaptodrosophila latifasciaeformis*)[13] (Supplementary Fig. 5). According to the BLASTp search, the amino acid sequence of ORF2 was homologous to the Wuhan insect virus 23 hypothetical protein (Fig. 2B; Supplementary Table 2), which is considered a capsid protein of a Partiti-like virus (found in an insect) because the virus was bipartite, and the protein sequence encoded by the other segment was homologous to RdRp[5]. Thus, we hypothesize that ORF2 is also a capsid protein of the Partiti-like virus. According to the BLASTn search, nucleotide sequences of ORF2 and ORF3 were homologous to small RNA sequences DmelPosVir_26 (e-value = $4 \times 10^{-25}$) and DmelPosVir_36 (e-value = $2 \times 10^{-98}$), respectively, which were derived from the aforementioned pool of wild-caught *Drosophila*[13]. This suggests that ORF2 and ORF3 constitute

viral sequences as well. Alternatively, ORF4 had no homologous sequences or conserved domains in the database. The four dsRNAs (dsRNA1–dsRNA4) had common nucleotide sequences at the 5′- and 3′-ends (AGAUUUUC and AGUCCC, respectively), although dsRNA4 had extra 8 nucleotides (GUUUUUU) at the 5′-end (Supplementary Fig. 6). Moreover, relative RNA titers between the four dsRNAs were mostly consistent (Supplementary Fig. 7). These data suggest that dsRNA1–dsRNA4 constitute segments of a previously undiscovered single Partiti-like virus, which we named as Drosophila biauraria male-killing partitivirus 1 (DbMKPV1). This notion was further supported by fluorescence in situ hybridization (FISH) using probes designed from each of the nucleotide sequences of dsRNA1–dsRNA4, which demonstrated colocalization of the four dsRNAs in the cytoplasm (Fig. 2D; Supplementary Fig. 8). Furthermore, the viral titers estimated by the amount of RdRp-encoding dsRNA1 were not significantly different between male and female embryos (Supplementary Fig. 9), precluding the causal relationship between the viral titers and sex-specific death.

## dsRNA4 is necessary for male-killing

Although DbMKPV1 is stably maintained in SP12F and tr.SP11-20, we noticed that certain viral segments could be lost during a few generations after injection, which might be due to the nature of Partitiviridae that each genome segment is packaged in a separate virus particle[12,14]. By utilizing this "unstable" condition, we analyzed the integrity of the four dsRNAs and the associated male-killing phenotype. Specifically, we injected the homogenate of SP12F adults into the

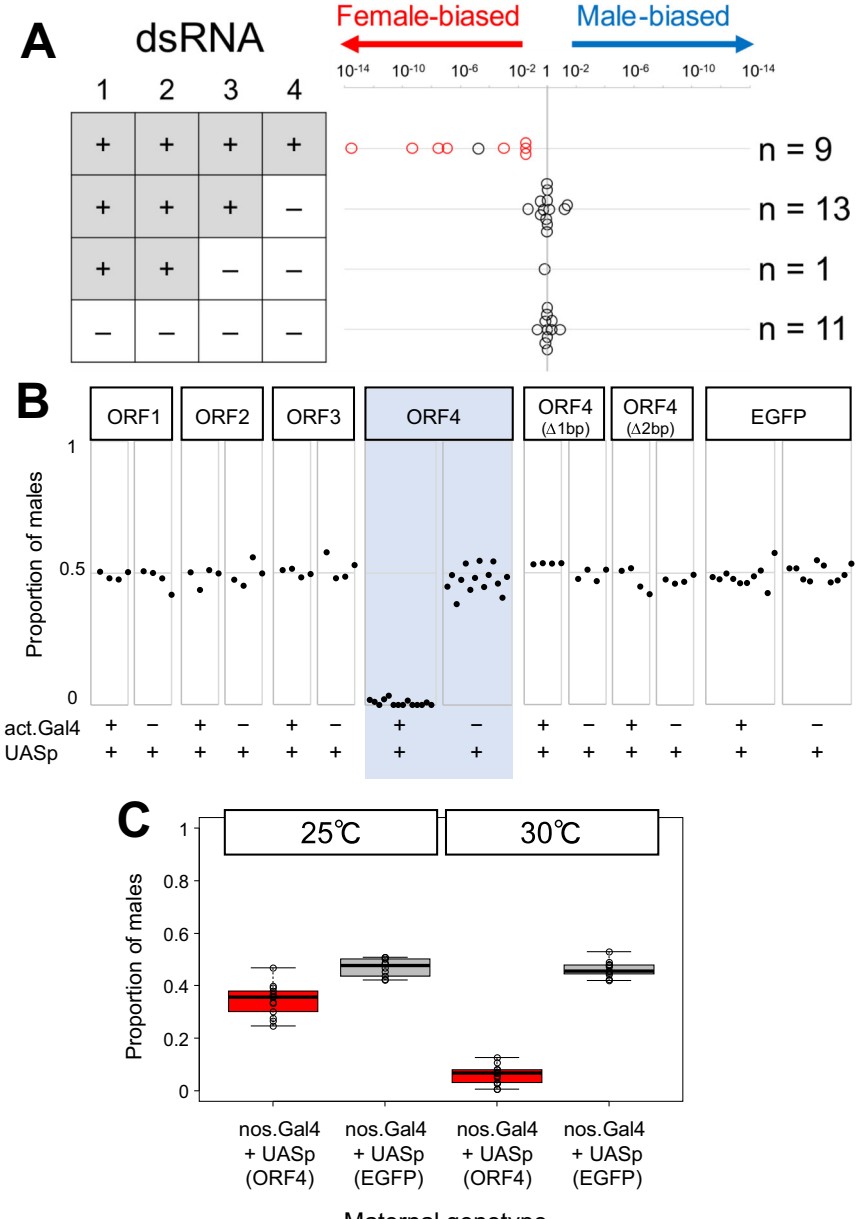

**Fig. 3 | Functional characterization of the four ORFs. A** Offspring sex ratios of 34 broods associated with the presence/absence of the four RNA fragments (dsRNA1–dsRNA4) in their mothers resulting from the transinfection of DbMKPV1 in *Drosophila biauraria*. **B** ORF overexpression effects on sex ratio in *Drosophila melanogaster* (using actin ubiquitous driver). ORF1, ORF2, ORF3, ORF4, two frameshifted ORF4, and EGFP were ectopically expressed. Male proportion among emerged adults is shown. **C** Effects of maternal expression of ORF4 using nanos driver on offspring sex ratio in *D. melanogaster*. Box plots represent the median, the first quartiles and third quartiles with whiskers drawn within the 1.5 IQR value. Independent ORF4-overexpressed broods (*n* = 11) and control broods (*n* = 10) were examined at 25 °C. Independent ORF4-overexpressed broods (*n* = 9) and control broods (*n* = 11) were examined at 30 °C. Male proportion in adults is shown. Raw data are provided as in Supplementary Data 4.

intersegmental membrane of the thorax of three SP11-20 female adults (Supplementary Fig. 10). The injected females produced offspring with female-biased sex ratios (23 females and 12 males; 14 females and 1 male; 13 females and 4 males). In the subsequent generations, brood sex ratios were segregated, ranging from strongly female-biased (46 females and 0 males) to slightly male-biased (11 females and 24 males). Next, the presence/absence of the four dsRNAs was examined in the mothers of 34 broods using RT-PCR (Fig. 3A). Among the 34 females examined, 9 females were positive for all the four dsRNAs, 13 females were positive only for dsRNA1, dsRNA2, and dsRNA3, 1 female was positive only for dsRNA1 and dsRNA2, and 11 females were negative for all dsRNAs. The presence of only dsRNA1 (encoding RdRp)

and dsRNA2 in one individual indicates that dsRNA3 and dsRNA4 are dispensable for virus multiplication/transmission. Notably, 14 females that lacked dsRNA4 produced offspring with sex ratios that were not strongly biased (Fig. 3A). These results suggest that dsRNA4 is necessary for male-killing.

## dsRNA4 is sufficient for male-killing

To confirm whether dsRNA4 is sufficient for male-killing, we utilized the GAL4/UAS system to overexpress ORF4 in *D. melanogaster*. Strikingly, ORF4 expression with the actin-GAL4 ubiquitous driver eliminated all or nearly all males (proportions of males: 0–5.63% [median: 0.93%]) without having adverse effects on the survival of female flies

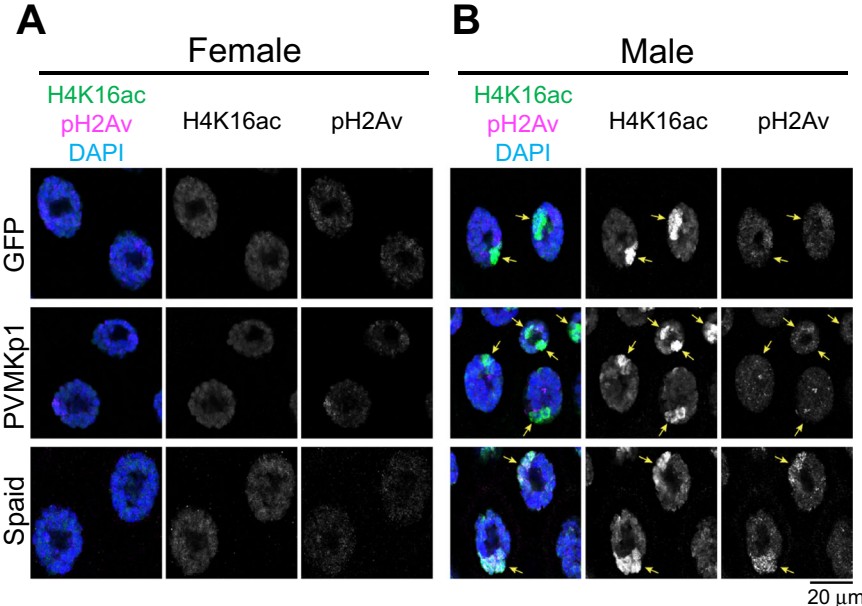

**Fig. 4 | Expression and the effect of two male-killing genes.** Female (**A**) and male (**B**) larval salivary glands of *D. melanogaster* expressing GFP (female, $n = 9$; male, $n = 10$), PVMKp1 (ORF4) (female, $n = 12$; male, $n = 9$), and Spaid (female, $n = 9$; male, $n = 9$). Staining for acetylated histone H4 lysine 16 (H4K16ac; green) indicates male X chromosomes. Staining for phosphorylated histone H2Av (pH2Av; magenta) indicates DNA damage. DAPI indicates DNA (blue). Arrowheads represent the expected location of male X chromosomes deduced from H4K16ac signals.

(Fig. 3B; Supplementary Fig. 11). Conversely, overexpression of ORF1, ORF2, or ORF3 did not affect the sex ratio (Fig. 3B). Moreover, the overexpression of frameshifted ORF4, generated by 1 bp or 2 bp deletions in the second codon (adjacent to ATG) of the ORF4-coding sequence, did not result in the male-killing phenotype (Fig. 3B) despite generating comparable mRNA levels to those of the intact ORF4 (Supplementary Fig. 12). This supports the notion that ORF4 exerts male-killing activity as a protein, which we termed partitivirus male-killing protein 1 (PVMKp1). When intact PVMKp1 was systemically (and zygotically) overexpressed in *D. melanogaster*, egg hatch rates were higher than 0.85 (Supplementary Fig. 13), which was higher than those observed in *D. biauraria* SP12F or tr.SP11-20 (see Fig. 1A), and 10.5–43.3% (median: 19.6%) of the third-instar larvae were males (Supplementary Fig. 14), which suggests that most males were not killed during embryogenesis, as is the case in *D. biauraria*, but were killed during larval and pupal development. Meanwhile, over-expression of PVMKp1 using nanos-GAL4 maternal driver also resulted in a male-killing phenotype in *D. melanogaster*, but the rate of killing was moderate (proportions of males in adults: 24.7–46.8% [median: 35.7%]). This was strengthened by GAL4 activation using a higher temperature (30 °C instead of the 25 °C standard) (proportions of males in adults: 0.47–12.7% [median: 6.58%]; Fig. 3C). The delayed/incomplete action of male-killing observed in *D. melanogaster* compared to that in the original host (*D. biauraria*) might be explained by the low PVMKp1 expression level. Delayed and incomplete male-killing phenotype because of the lower titer of male-killers has been described in *Drosophila* flies and *Hypolimnas* butterflies[15–17]. A second scenario is that the overexpression timing by the zygotic and maternal drivers may be nonoptimal (too late/early) for inducing complete male-killing.

## Mechanistic insight into male-killing

Next, we sought to obtain mechanistic insight into how PVMKp1 induces selective death of males. The mechanism of male-killing has been well described in the association of *Drosophila* with its bacterial symbionts[18]. In particular, a bacterial male-killing toxin Spaid was recently identified in the male-killing *Spiroplasma* from *D. melanogaster*[19]. Its direct binding causes DNA damage on the male X chromosome[19], which was readily reproduced through overexpression in the larval salivary gland cells (Fig. 4). In striking contrast, no obvious signs of DNA damage (i.e., phosphorylation of histone H2Av) were detected by the overexpression of PVMKp1 (Fig. 4). These results imply a difference in mechanisms associated with Spaid and PVMKp1 despite the phenotypic resemblance at the macroscopic level. However, we cannot rule out the possibility that weak DNA damage below the detection limits is responsible for the lethality caused by PVMKp1.

## Evolutionary insight into male-killing viruses

Recently, Fujita et al.[8] discovered a virus (Osugoroshi virus; OGV) belonging to the family Partitiviridae, from a male-killing moth matri-line, *Homona magnanima*[20]. Although the OGV is considered to cause male-killing and comprises 27 RNA segments (3 of which were RdRp), the segments responsible for male-killing have not been identified. Our BLAST search suggested that none of the protein sequences derived from the 27 RNA segments were homologous to PVMKp1 at the amino acid level. We hypothesize that DbMKPV1 and OGV have indepen-dently acquired male-killing genes through shuffling of genomic seg-ments, or reassortment, which is quite common in segmented viruses[21]. The fact that a single protein encoded by a viral RNA segment induces male-killing imply that other Partiti-like viruses, and possibly segmented viruses of other taxa as well, can easily acquire such gene to manipulate host reproduction and cause male-killing in a wide variety of arthropods.

## Methods

### Collection and maintenance of *Drosophila biauraria*

We used laboratory stocks of *Drosophila biauraria* (Diptera; Droso-philidae), which were originally collected at the Field Science Center for Northern Biosphere, Hokkaido University located at Tomakomai, Hokkaido in 2011 and 2015 using standard banana traps and sweeping[22]. Females were brought into the lab and iso-female lines (SP11-20, TM15-12, TM15-22, TM15-41, and TM15-47) were established using banana medium (270 g of bananas, 68 g of malt, 68 g of molas-ses, 40 g of dried yeast, 20 g of agar, and 2 liters of distilled water,

supplemented with 9 ml of propionic acid and 0.036% (w/v) of butyl p-hydroxybenzoate). The all-female line (SP12F) was maintained by mating with males of a standard iso-female line (SP11-20). *D. biauraria* were maintained at 20 °C with banana medium under 16 h; 8 h light-dark regimen.

## Estimate of egg hatch rates

Embryos were collected from grape-juice agar plates and dechorionated in 2.8% sodium hypochlorite solution. They were subsequently soaked in phosphate-buffered saline (PBS) containing 0.1% Triton X-100 (PBT), placed at 25 °C for 4 d, and counted under dissecting microscope. Hatched larvae (first-instar larvae) and unhatched but well-developed embryos were rinsed with 99% ethanol three times and stored in 99% ethanol.

## Genetic sexing of embryos and hatchlings of *D. biauraria*

Y chromosome-linked markers for *D. biauraria* are unable to obtain by inferring the genome data of *D. melanogaster* because orthologs of Y chromosome-linked sequences of *D. melanogaster* are present in both males and females in the *montium* group, such as *D. biauraria*[23]. Therefore, by comparing the genomic data of male and female *D. biauraria* obtained by Illumina HiSeq, we isolated male-specific sequences. A pair of primers, DbY_c52202_F2 and DbY_c52202_R2 (Supplementary Table 1), designed from one of the sequences discriminated males and females unambiguously by polymerase chain reaction (PCR). A pair of primers, Db-actin5C-68-F and Db-actin5C-68-R (Supplementary Table 1), that amplify *actin-5C* was used to confirm that *D. biauraria* genomic DNA was properly extracted.

Each of the embryos and first-instar larvae, picked up from ethanol, was briefly air-dried and squashed in 20 μL of PrepMan™ Ultra Sample Preparation Reagent (ThermoFisher, Cat. No. 4318930). Samples were then incubated at 100 °C for 10 min, vortexed for 15 s and centrifuged at 20,000 × *g* for 2 min before being subjected to PCR.

## Transinfection

After homogenizing 80 adults of SP12F in 300 μL of PBS, supernatant was sterilized using a 0.22-μm filter (MILLEX GV, Merck Millipore, Cat. No. SLGV033RS) and 0.2 μL was injected into the intersegmental membrane of thorax of mated females of *D. biauraria* SP11-20 line using a glass capillary needle (Drummond, Cat. No. 2-000-005) and an air pump Linicon LV-125 (Nitto Kohki Co., Ltd.). Injected females were individually placed in the vials with the banana medium to produce offspring. In subsequent generations, each female adult was placed with an SP11-20 male in a vial with the banana medium. An all-female line resulted from the transinfection, referred to as tr.SP11-20, was maintained by mating with males of SP11-20.

## RNA sequencing

Total RNA was extracted from 30 individuals from each of the seven *D. biauraria* lines (30 females from SP12F and tr.SP11-20; 15 females and 15 males from SP11-20, TM15-12, TM15-22, TM15-41, and TM15-47) using RNeasy (Qiagen, Germany). Extracted RNA was subjected to RNA sequencing using Illumina HiSeq 2500 following the removal of ribosomal RNA using Ribo-Zero rRNA Depletion Kit (Illumina, Cat. No. MRZH11124) by Macrogen (South Korea). Generated raw RNA-seq reads of the seven lines were cleaned by Trimmomatic version 0.36[24]. A reference transcriptome assembly was constructed by de novo assembly of merged clean reads of three lines (SP12F, tr.SP11-20, and SP11-20) using Trinity version 2.2.0[25]. The assembled contigs were annotated with descriptions of top-hit proteins in the NCBI non-redundant (NCBI-nr) protein database by blastx search (e-value <10$^{-3}$). The coding DNA sequence (CDS) regions of the contigs were estimated using TransDecoder version 3.0.1 (https://github.com/TransDecoder/TransDecoder) and annotated with descriptions of hit protein domains in the Pfam database by HMMER version 3.1b2[26]. Expression levels (transcripts per million (TPM) values) of the assembled contigs were calculated for each line by mapping the clean reads to the reference transcriptome assembly using "align_and_estimate_abundance.pl" script bundled with Trinity which uses Bowtie2 version 2.2.6[27] and RSEM version 1.2.31[28]. Contigs highly specifically expressed in the two all-female matrilines (SP12F and tr.SP11-20) were extracted by filtering with TPM value > 100 and fold change > 10 for the two lines when compared with other lines.

## Rapid amplification of 3′-cDNA end (3′-RACE)

Total RNA extracted from *D. biauraria* SP12F was used as a template for RACE using ALL-TAIL™ Kit (Bioo Scientific Corporation, Cat. No. 5205). Following the manufacturer's protocol, AIR™ Adenylated Linker C was ligated to the total RNA, followed by the reverse transcription using Linker C Universal RT-PCR Primer, a primer complementary to the AIR Adenylated Linker C. The products were subjected to PCR amplification using a primer specific to one of the four contigs (Supplementary Table 1) and Linker C Universal RT-PCR Primer. PCR products electrophoresed on agarose gel were excised, purified by Wizard® SV Gel and PCR Clean-Up System (Promega, Cat. No. A9281) and subjected to direct Sanger sequencing using ABI 3730XL sequencer (Applied Biosystems).

## Phylogenetic analyses

One hundred thirteen putative amino acid sequences of RNA-dependent RNA polymerase (RdRp) of Partitiviridae, including an ORF1 (dsRNA1) sequence identified in *Drosophila biauraria* were aligned using the MAFFT version 7 using the "-auto" setting[29]. Multiple-sequence alignments were trimmed with the trimAL version 1.3 using setting "-automated1" to remove uninformative columns[30]. The phylogenetic analysis was performed using raxmlGUI 2.0 version 2.0.7[31]. The BIC-based best model according to ModelTest-NG version 0.1.7[32] was used for phylogenetic tree reconstruction using ML + thorough bootstrap with a replication setting "autoMRE"[33].

## Fluorescent in situ hybridization (FISH)

Localization of the four dsRNAs of DbMKPV1 were visualized by whole-mount fluorescence in situ hybridization (FISH) using the midgut of female adults of 13 d after emergence as described previously[34,35], using the fluorochrome-labeled oligonucleotide probes listed in Supplementary Table 1. AlexaFluor 488-labeled probes were used for the detection of dsRNA1 and dsRNA4. Alexa-Fluor 647-labeled probes were used for the detection of dsRNA2. AlexaFluor 555-labeled probes were used for the detection of dsRNA3. Designing probes were conducted by using Stellaris Probe Designer version 4.2 (https://www.biosearchtech.com/stellarisdesigner/, Biosearch Technologies) for picking up the plurality of candidate sequences. Then specificities of the probes were checked using ProbeCheck (http://131.130.66.200/cgi-bin/probecheck/probecheck.pl) and BLASTN 2.6.1 (https://blast.ncbi.nlm.nih.gov/Blast.cgi), and narrowed down to the 2 probes per ORF. Host cell nuclei were counterstained with 4′,6-diamidino-2-phenylindole (DAPI). Observations were made using a laser scanning confocal microscope LSM710 (Carl Zeiss, Germany) and analyzed using ZEN 2009 software (Carl Zeiss, Germany). The specificity of in situ hybridization was confirmed by the following control experiments: a no-probe control, an RNase digestion control as described[36].

## *Drosophila melanogaster* fly stocks and genetics

Laboratory stocks of *D. melanogaster* were maintained at 25 °C with banana medium. The following lines were obtained from the Bloomington *Drosophila* Stock Center (BDSC) at Indiana University and the Department of *Drosophila* Genomics and Genetic Resources (DGGR) at Kyoto Institute of Technology: *Actin5C-GAL4* (*act-GAL4*; BDSC #4414), *nanos-GAL4::VP16* (*nos-GAL4*; BDSC #4937), *UASp-EGFP* (DGGR #116072), and *CyO, ActGFP* (the green balancer; DGGR #107783). The

four ORFs (ORF1–ORF4) encoded in the DbMKPV1 genome were expressed by the GAL4/UAS system[37] in *D. melanogaster*. For the zygotic expression, *Actin5C-GAL4/CyO* flies were crossed to homozygous *UAS* transgenic flies. For maternal expression[38], *nanos-GAL4* females were crossed to *UAS* males, and the resultant female progeny were mated with Oregon-R (OR-NIG) males. For the GAL4/UAS expression in larval salivary glands (Fig. 3), we used a recombined *Actin5C-GAL4, tubulin-GAL80ts* line (generated by a combination of BDSC #4414 and BDSC #7108) balanced with *CyO, ActGFP* (DGRC #107783) to avoid male lethality and obtain wandering third-instar larvae. After maintained at 20 °C for 7–8 d, crosses were shifted to 29 °C and kept for 1 d before dissection. GFP on the balancer chromosome was utilized as a selection marker. For the expression of Spaid, *UASp-Spaid-GFP* transgenic flies were used.

### Construction of transgenic *D. melanogaster* lines

First, total RNA was extracted from infected whole *D. biauraria* adult females (SP12F, $n = 30$) using the RNeasy Mini kit (Qiagen). Total RNA was used for cDNA synthesis by the PrimeScript RT-PCR Kit (Takara Bio, Japan). The four DbMKPV1 ORFs (without in-frame stop codons) were PCR amplified from the synthesized cDNA with specific primers. To generate two frameshift mutations of ORF4 (Δ1bp and Δ2bp), we designed and used two forward PCR primers with one/two-nucleotide deletion in the second codon of the coding sequence (5′-ATG GCG CAT-3′). Primers used for PCR amplification are listed in Supplementary Table 1. Each PCR fragment was cloned into the pENTR vector by the pENTR/D-TOPO cloning kit (Thermo Fisher Scientific). We utilized PrimeSTAR MAX DNA Polymerase (Takara Bio) for all PCR reactions above. The Gateway cassette containing the ORF fragments was transferred into the pPW destination vector (The *Drosophila* Genomics Resource Center #1130; The *Drosophila* Gateway Vector Collection by Terence Murphy) by the LR clonase II enzyme mix kit (Thermo Fisher Scientific) to construct pUASp-ORFs plasmids. Transgenic fly lines were generated by the standard microinjection method for P-element transformation (BestGene).

### Reverse transcription quantitative polymerase chain reaction (RT-qPCR)

Total RNA was extracted from each of the three *D. melanogaster* adults from respective genotypes (act-Gal4.UAS-ORF4$^{\Delta1.M7}$, CyO.UAS-ORF4$^{\Delta1.M7}$, act-Gal4.UAS-ORF4$^{\Delta1.M3}$, CyO.UAS-ORF4$^{\Delta1.M3}$, act-Gal4.UAS-ORF4$^{\Delta2.M3}$, CyO.UAS-ORF4$^{\Delta2.M3}$, act-Gal4.UAS-ORF4$^{F6}$, CyO.UAS-ORF4$^{F6}$, act-Gal4.UAS-ORF4$^{M1}$, and CyO.UAS-ORF4$^{M1}$), using RNeasy Plus Mini kit (Qiagen). cDNA was synthesized from the total RNA by PrimeScript™ RT reagent Kit with gDNA Eraser (Perfect Real Time) (Takara Bio, Japan). qPCR was conducted by using KOD SYBR® qPCR Mix (TOYOBO, Japan) using two primer sets of ORF4_q_F1 and ORF4_q_R1 (Supplementary Table 1) for dsRNA4, and Dm_rp49_q_F1 and Dm_rp49_q_R1 (Supplementary Table 1) for DmRp49 as reference gene. Thermal conditions were managed by LightCycler® 96 Instrument (Roche, Switzerland) as 95 °C for 300 s followed by 40 cycles of 95 °C for 5 s and 60 °C for 20 s. The relative amounts of the RNA were calculated using quantification cycles (Cq) by ddCq analysis. The RNA levels of dsRNA4 were normalized by those of DmRp49, and the relative RNA levels of dsRNA4 were estimated by setting one of the act-Gal4.UAS-ORF4$^{F6}$ as 1. Expression of GFP was checked by a fluorescent microscope MZ10F (Leica).

### *D. melanogaster* staining and imaging

Larval salivary glands were dissected out from wandering third-instar larvae and fixed in 4% paraformaldehyde (EM Grade; Electron Microscopy Sciences, 15710) diluted in PBS for 20 min at room temperature. After washed in PBT and treated with a blocking buffer [PBT containing 1% bovine serum albumin (BSA, heat shock fraction; Sigma-Aldrich, A7906)] for 30 min, tissues were incubated with primary antibodies at

4 °C overnight, then washed in PBT and incubated with secondary antibodies at room temperature for 90 min. Antibodies were diluted in the blocking buffer. The following primary antibodies were used: rabbit anti-acetyl-histone H4 lysine 16 (H4K16ac; 1/2,000; Upstate, Sigma-Aldrich 07-329), mouse anti-phospho-histone H2Av (pH2Av; 1/500; DSHB, UNC93-5.2.1). Secondary antibodies (1:2,000; Alexa Fluor Plus 555/647 conjugate) were purchased from Thermo Fisher Scientific (A32773 and A32795). DNA staining was carried out with DAPI (0.5 µg/mL; Nacalai Tesque, 19178-91) together with secondary antibodies. Stained tissues were washed in PBT, mounted in ProLong Glass Antifade Mountant (Thermo Fisher Scientific, P36980), and observed under a confocal microscope (Olympus FLUOVIEW FV3000) equipped with a 40×/1.4 oil immersion objective (2×zoom scan; frame size: 1024 × 1024; 0.42 µm/slice with optimal intervals). The brightness and contrast of a single optical section were adjusted uniformly on the entire images and only black/white input levels were modified using EBIImage package version 4.5.22 under R version 4.1.2 and Fiji ImageJ version 1.51r.

### Statistics and reproducibility

The number of analyzed individuals and the number of biological replicates (at least thrice) are indicated in the figure legends. For each gene used for constructing transgenic flies, two independently established fly lines were examined.

### Reporting summary

Further information on research design is available in the Nature Portfolio Reporting Summary linked to this article.

### Data availability

RNA-seq raw data generated in this study have been deposited to DDBJ Sequence Read Archive (DRA) under the accession number DRA011109. RNA-seq contig data are deposited in https://doi.org/10.6084/m9.figshare.22047350.v1. All other data are available in the main text or the supplementary materials with raw data underlying the figures provided as Supplementary Data 1–7. The biological materials used in this study are available from authors.

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

## Acknowledgements

The authors thank Greg Hurst for helpful comments on the manuscript. We also thank Manabu Ote for helpful discussion, Masae Takashima and Takuhiko Yokoyama for technical assistance, and Syusuke Egoshi, Kosuke Dodo, Mikiko Sodeoka, and Minoru Yoshida for allowing us to use LSM710 confocal microscope. Fly stocks provided by Kyoichi Sawamura, the Bloomington Drosophila Stock Center (NIH P40OD018537), and KYOTO Stock Center (DGRC) in Kyoto Institute of Technology were used in this study. This work was supported by Japan Society for the Promotion of Science (15K07189 and 18K06383 to M.W.), the Hakubi Project of Kyoto University (to T.H.), JST ERATO Grant Number JPMJER1902 (to T.H.), Nagase Science and Technology Foundation (to T.H.), and Cabinet Office, Government of Japan, Moonshot Research and Development Program for Agriculture, Forestry, and Fisheries (funding agency: Bio-oriented Tech-nology Research Advancement Institution) (no. JPJ009237 to D.K.).

## Author contributions

D.K. and M.W. supervised the study. D.K., T.K.K., and M.W. collected or reared the samples. D.K., A.F., T.H., K.N., and M.W. generated data through sample preparation and/or laboratory work. D.K., M.T., A.J., and T.N.S analyzed and/or curated data. D.K. and M.W. wrote the paper with input from all authors.

## Competing interests

The authors declare no competing interests.
