## [Peer Review File · Nature Communications]

A male-killing gene encoded by a symbiotic virus of *Drosophila*REVIEWER COMMENTS

Reviewer #1 (Remarks to the Author):

Review for Kageyama et al., entitled "A male-killing gene encoded by a symbiotic virus of *Drosophila*", submitted to Nature Communications

In this study, the authors have identified a lineage of *Drosophila biauraria* with strong sex-ratio bias towards females. Through a series of experiments they identified a novel partitivirus infecting these flies, and demonstrated that injection of this virus into a different fly line resulted in a maternally transmitted all-female trait. The authors then used the natural phenomenon of partitivirus segment loss that can sometimes occur during transmission to show that loss of one of the virus segments correlated with loss of the female-bias phenotype. They introduced all four ORFs from four segments of the virus genome into *Drosophila melanogaster*, and only the ORF on the candidate virus segment (ORF4) caused female bias when overexpressed. The mechanism of male-killing was compared with the mechanism known from male-killing bacterial symbionts, and it was revealed that the partitivirus mechanism of male-killing does not involve DNA damage to the Y chromosome, as it does for the Spaid gene product produced by *Spiroplasma* symbionts.

These findings are significant because the authors have identified the specific gene involved in female bias that is encoded by a male-killing virus. This finding is novel and adequately supported by the data provided in the manuscript.

Minor comments:

If it were possible I would have liked to see more evidence that the dsRNA4 truly belonged to the other dsRNAs in the partitivirus. In figure S8 I would have liked to see the presence or absence of the partitivirus segments in the first three generations shown in the pedigree, especially given the variability in the male-killing trait shown for some fly mothers. Would it be possible to screen field-caught individuals or do lab-based experiments with increased sample sizes to demonstrate a high correlation between the presence of dsRNA4 and the remaining viral segments (particularly dsRNA1, which clearly belongs to the partitiviruses). Is dsRNA4 ever detected without dsRNA1 also being detected? This is important because the novelty of the findings rests on the assumption that ORF4, the male-killing gene, is from a maternally inherited virus.

The claim that ORF2 encodes a capsid protein is too strong on line 146 based upon the evidence. If the authors wanted to demonstrate the function of ORF2, perhaps they could provide TEM images of transgenic *D. melanogaster* overexpressing ORF2 protein. This is a minor point though for this manuscript, and perhaps it would be better to just state that dsRNA1 and dsRNA2 appear essential for the virus.

The images in Figure S7 may need to be brightened. I had trouble seeing the signal for dsRNA2 and dsRNA3 in panel A even on my computer screen.

Reviewer #2 (Remarks to the Author):

Summary: This is a beautiful and thorough study that is clearly and succinctly described. The identification of the causal protein is elegantly shown, and has implications to virology, genetics, evolution, and applied biology. There are many interesting avenues for future research, such as determination of the efficacy range of the protein and its mode of action, screening for similar viruses and the dsRNA4 gene in other insects, et cetera. There is really nothing more to say. Well done and important.

(Minor) Specific Comments

L114 – In what insect was the Wuhan insect virus 23 containing the sequence similar to ORF2?

L130 – Did you check the dead or dying male embryos by FISH and also by quantification, to determine whether the virus was over-replicating in these embryos compared to controls? This would be a good place to mention this.

L131 – Using the unstable transmission of different “chromosomes” of the segmental virus is a very nice trick for investigating the causal genes.

L149 – And then using transformation to confirm the role of dsRNA4 is icing on the cake. Very nicely done.

L190 – I draw the author’s attention to another paper showing a sex ratio effect of an RNA virus, although it is not a male killer, and implication of a particular gene.

Wang, F., Fang, Q., Wang, B., Yan, Z., Hong, J., Bao, Y., ... & Ye, G. (2017). A novel negative-stranded RNA virus mediates sex ratio in its parasitoid host. *PLoS pathogens*, 13(3), e1006201.

Point to Point Response to the Reviewers

Manuscript ID: NCOMMS-22-34698-T

We are grateful to the editor and reviewers for taking their precious time for our manuscript. We sincerely appreciate their valuable and insightful comments. We have revised and improved our manuscript as suggested. In the following, we provide a point-to-point response to every question raised. We have highlighted portions of the manuscript that have been updated in response to the reviewers' comments.

Reviewer #1 (Remarks to the Author):

Review for Kageyama et al., entitled "A male-killing gene encoded by a symbiotic virus of Drosophila", submitted to Nature Communications

In this study, the authors have identified a lineage of Drosophila biauraria with strong sex-ratio bias towards females. Through a series of experiments they identified a novel partitivirus infecting these flies, and demonstrated that injection of this virus into a different fly line resulted in a maternally transmitted all-female trait. The authors then used the natural phenomenon of partitivirus segment loss that can sometimes occur during transmission to show that loss of one of the virus segments correlated with loss of the female-bias phenotype. They introduced all four ORFs from four segments of the virus genome into Drosophila melanogaster, and only the ORF on the candidate virus segment (ORF4) caused female bias when overexpressed. The mechanism of male-killing was compared with the mechanism known from male-killing bacterial symbionts, and it was revealed that the partitivirus mechanism of male-killing does not involve DNA damage to the Y chromosome, as it does for the Spaid gene product produced by Spiroplasma symbionts.

These findings are significant because the authors gave identified the specific gene involved in female bias that is encoded by a male-killing virus. This finding is novel and adequately supported by the data provided in the manuscript.

Response: We thank the reviewer for the positive comments on the work.

Minor comments:

If it were possible I would have liked to see more evidence that the dsRNA4 truly belonged to the other dsRNAs in the partitivirus. In figure S8 I would have liked to see the presence or absence of the partitivirus segments in the first three generations shown in the pedigree, especially given the variability in the male-killing trait shown for some fly mothers. Would it be possible to screen field-caught individuals or do lab-based experiments with increased sample sizes to demonstrate a high correlation between the presence of dsRNA4

and the remaining viral segments (particularly dsRNA1, which clearly belongs to the partitiviruses). Is dsRNA4 ever detected without dsRNA1 also being detected? This is important because the novelty of the findings rests on the assumption that ORF4, the male-killing gene, is from a maternally inherited virus.

Response: We also consider that it is important to verify that dsRNA4 is a component of the virus. Unfortunately, samples for the first three generations of the Fig. S10 (S8 of the previous version) no longer exist. Therefore, in response to the above comment, we have examined the titres of dsRNA1–dsRNA4 in adults of SP12F and tr.SP11-20 lines that are maintained in the lab. Clearly, the relative RNA titres between the four dsRNAs were mostly consistent although virus titres were variable between individuals (Fig. S7). This would be a good information that support the integrity of the four dsRNAs in addition to the shared sequence in 5'-end and 3'-end (typical feature of a single segmented virus) (Fig. S6) and the colocalization of the four RNAs (Fig. S8). Therefore, we are more confident than ever that dsRNA4 is a component of the virus.

The claim that ORF2 encodes a capsid protein is too strong on line 146 based upon the evidence. If the authors wanted to demonstrate the function of ORF2, perhaps they could provide TEM images of transgenic D. melanogaster overexpressing ORF2 protein. This is a minor point though for this manuscript, and perhaps it would be better to just state that dsRNA1 and dsRNA2 appear essential for the virus.

Response: We agree with this. Following this comment, we deleted this part (lines 149–150).

The images in Figure S7 may need to be brightened. I had trouble seeing the signal for dsRNA2 and dsRNA3 in panel A even on my computer screen.

Response: Following this suggestion, we have brightened the Fig. S8 (S7 of the previous version) as a whole.

Reviewer #2 (Remarks to the Author):

Summary: This is a beautiful and thorough study that is clearly and succinctly described. The identification of the causal protein is elegantly shown, and has implications to virology, genetics, evolution, and applied biology. There are many interesting avenues for future research, such as determination of the efficacy range of the protein and its mode of action, screening for similar viruses and the dsRNA4 gene in other insects, et cetera. There is really nothing more to say. Well done and important.

(Minor) Specific Comments

L114 – In what insect was the Wuhan insect virus 23 containing the sequence similar to ORF2?

Response: Wuhan insect virus 23 is from mosquitoes. The host information was included in Table S2. Moreover, all the hosts of the viruses used in phylogenetic analyses, are shown in Dataset S3.

L130 – Did you check the dead or dying male embryos by FISH and also by quantification, to determine whether the virus was over-replicating in these embryos compared to controls? This would be a good place to mention this.

Response: Following this comment, we newly quantified the titres of viral RNA by RT-qPCR and found that these were not different between male and female embryos. These data were presented as Fig. S9 and mentioned in the lines 131–133.

L131 – Using the unstable transmission of different “chromosomes” of the segmental virus is a very nice trick for investigating the causal genes.

L149 – And then using transformation to confirm the role of dsRNA4 is icing on the cake. Very nicely done.

L190 – I draw the author’s attention to another paper showing a sex ratio effect of an RNA virus, although it is not a male killer, and implication of a particular gene.

Wang, F., Fang, Q., Wang, B., Yan, Z., Hong, J., Bao, Y., ... & Ye, G. (2017). A novel negative-stranded RNA virus mediates sex ratio in its parasitoid host. PLoS pathogens, 13(3), e1006201.

Response: We thank the reviewer for these very positive comments on the work and for literature information.

REVIEWERS' COMMENTS

Reviewer #1 (Remarks to the Author):

The authors have satisfied my minor concerns and those of the other reviewer for this manuscript. This is a nice manuscript and I look forward to seeing its publication.

Reviewer #2 (Remarks to the Author):

I have reviewed the revised manuscript and I am satisfied with the changes. The work in this paper is very thoroughly done, and provides significant contributions to the literature on viruses that distort the reproduction of their hosts.